# Impact of Gut Dysbiosis on the Risk of Non-Small-Cell Lung Cancer

**DOI:** 10.3390/ijerph192315991

**Published:** 2022-11-30

**Authors:** Yu-Feng Wei, Ming-Shyan Huang, Cheng-Hsieh Huang, Yao-Tsung Yeh, Chih-Hsin Hung

**Affiliations:** 1Institute of Biotechnology and Chemical Engineering, I-Shou University, Kaohsiung 84001, Taiwan; 2School of Medicine for International Students, College of Medicine, I-Shou University, Kaohsiung 82445, Taiwan; 3Department of Internal Medicine, E-Da Cancer Hospital, Kaohsiung 82445, Taiwan; 4PhD Program in Environmental and Occupational Medicine, Kaohsiung Medical University, Kaohsiung 80708, Taiwan; 5Aging and Disease Prevention Research Center, Fooyin University, Kaohsiung 83102, Taiwan; 6Department of Medical Laboratory Sciences and Biotechnology, Fooyin University, Kaohsiung 83102, Taiwan

**Keywords:** dysbiosis, gut microbiota, microbiome, non-small-cell lung cancer

## Abstract

Background: The imbalance of gut microbiota, dysbiosis, is associated with various malignant diseases. This study aimed to identify the characteristics of gut microbiota in age-matched treatment-naïve non-small-cell lung cancer (NSCLC) patients and healthy individuals to investigate possible gut-microbe-related pathways involved in the development of NSCLC. Methods: We enrolled 34 age-matched NSCLC patients and 268 healthy individuals. Hypervariable V3–V4 amplicons of 16S rRNA in freshly collected fecal samples were sequenced. Diversity, microbial composition, functional pathways, smoking history, and gut-microbe-related comorbidities were analyzed to assess the factors associated with the risk of NSCLC. Results: Microbial alpha diversity was decreased in the patients with NSCLC, and beta diversity was significantly different between the patients and controls (*p* < 0.001). After adjustments for sex, smoking history, hypertension, diabetes mellitus, chronic obstructive pulmonary disease, and 11 abundant microbes with significant differences between the patients and controls, the enrichment of *Anaerotruncus* spp. and *Bacteroides caccae* was associated with an increased risk of NSCLC (*p* = 0.003 and 0.007, respectively). The areas under receiver operating characteristic curves were 71.4% and 66.9% for *Anaerotruncus* spp. and *Bacteroides caccae*, respectively (both *p* < 0.001). Furthermore, the abundance of *Bacteroides caccae* was positively correlated with steroid hormone biosynthesis (*p* < 0.001), N-glycan biosynthesis (*p* = 0.023), glycosaminoglycan degradation (*p* < 0.001), lipoic acid metabolism (*p* = 0.039), peroxisome (*p* < 0.001), and apoptosis (*p* < 0.001), but inversely related to glycerolipid metabolism (*p* < 0.001). *Anaerotruncus* spp. was positively associated with decreased biosynthesis of ansamycin only (*p* = 0.001). No overlapping signaling pathways were modulated by *Bacteroides caccae* or *Anaerotruncus* spp. Conclusions: Our results revealed that fecal *Anaerotruncus* spp. and *Bacteroides caccae* were abundant and may be associated with the risk of NSCLC regardless of sex, smoking history, and gut-microbe-related comorbidities. Further investigations on the mechanism underlying the potential association between gut dysbiosis and the development of NSCLC are warranted.

## 1. Introduction

Lung cancer remains the leading cause of cancer deaths in most countries, including Taiwan [1]. Non-small-cell lung cancer (NSCLC) is the most common histological type, accounting for 85% of all lung cancer cases. More than 70% of patients with NSCLC present with locally advanced or metastatic disease (Stage III or IV). Despite advances in lung cancer treatment, including targeted therapy and immunotherapy, the overall prognosis is still poor, with a median 5-year overall survival rate of only 25%, and lower than 10% in patients with metastatic disease [2]. Understanding the risk factors and pathways associated with NSCLC is crucial to make an early diagnosis and improve treatment strategies and outcomes. 

Emerging evidence has shown associations between microbial dysbiosis and the pathogenesis of various diseases, including cancers and common chronic diseases [3,4]. Gut microbiota may contribute to a shift in the human host microbiome, thereby modulating immuno-inflammatory responses and the development of diseases [5]. Previous studies have demonstrated that several microbiota subpopulations can expand via pathological dysbiosis and that this can affect the production of bacteriotoxins, genotoxicity, and a virulence effect to trigger both inflammation and tumorigenesis [4]. A previous animal study established a link between microbiota–immune crosstalk and lung cancer development [6], and a cohort study showed that the increased use of antibiotics was associated with an increase in lung cancer incidence [7]. Despite extensive evidence linking gut microbiota with lung diseases [8,9,10,11], the spectrum of gut microbiota related to the risk of lung cancer remains largely unknown. 

In the setting of lung cancer, most studies have focused on the impact of lung microbes because of their direct contact. However, the association between the gut microbiome and lung cancer is increasingly being explored. The enrichment of *Enterococcus* spp. and decreased abundances of *Bifidobacterium* spp. and *Actinobacteria* spp. have been associated with lung cancer. Furthermore, functional impairment of the gut microbiome has been shown to contribute to the progression of lung cancer [12]. The gut microbiota has also been shown to modulate responses to immunotherapy in lung cancer and possibly to serve as a predictor of immunotherapy outcomes [13,14,15]. In addition, differences in microbial composition have been associated with smoking [16,17], a well-established lung cancer risk factor [18]. Intriguingly, tobacco smoking has been associated with increased microbial diversity [19]. In general, microbial diversity is decreased in patients with diseases. For example, decreased microbial diversity has been associated with reduced lung function in patients with cystic fibrosis [20]. Because of the inconsistent results in previous studies, further investigations into the role of gut microbiota in the development, progression, and treatment of lung cancer are warranted.

Accordingly, the purpose of this study was to identify and compare the core microbes in the gut between treatment-naïve NSCLC patients and age-matched healthy individuals. In addition, we investigated the associations and potential pathways through which gut microbes may contribute to the development of NSCLC. 

## 2. Materials and Methods

### 2.1. Subjects and Sample Collection

Thirty-four patients diagnosed with NSCLC were recruited from September 2015 to July 2016 at E-Da Cancer Hospital. Clinical data of all NSCLC patients were recorded including age, sex, smoking status, cancer staging at diagnosis, epidermal growth factor receptor (EGFR) mutation status and subtype, and comorbid diseases at baseline. The patients who had a history of antibiotic use as well as the consumption of probiotics, prebiotics, or symbiotics in the previous month were excluded. We randomly selected 268 healthy individuals (age-matched controls: 64.1 ± 5.9 years; males, *n* = 113; females, *n* = 155) with normal chest radiographs as the control group from 1491 people who participated in health examinations in 2018, and those who had known diseases or medical records that may have affected gut microbiota composition (i.e., type II diabetes [1], hypertension [2], and cardiovascular diseases [3]) were excluded. The protocol and procedures of the current study were reviewed and approved by the Institutional Review Boards (IRBs) of Fooyin University Hospital (IRB number: FYH-IRB-107-03-01) and E-Da Hospital (IRB number: EMRP36107N). Informed consent was obtained from all participants. Fecal samples were collected using a standard collection kit (Cat. No. 21250. Iron Will Biomedical Technology, Taiwan) with stool DNA stabilizer (SKU: 1038111100, Invitek, Berlin, Germany) and preserved at −80 °C before further analysis.

### 2.2. DNA Extraction, Polymerase Chain Reaction (PCR), and Targeting Sequencing

The DNA contents of feces collected from the NSCLC patients and healthy controls were extracted using a Qiagen stool DNA kit (QIAmp DNA Stool Mini Kit, Hilden, Germany) according to the manufacturer’s instructions. DNA samples with optical density (OD) 260/280 nm in the range of 1.8–2.0 were stored at −20 °C before targeting sequence analysis. The V3 and V4 regions of 16S rDNA were amplified with bacterial-specific primers [21]. The primer sequences were: Forward (5′-TCG TCG GCA GCG TCA GAT GTG TAT AAG AGA CAG CCT ACG GGN GGC WGC AG-3′) and Reverse (5′-GTC TCG TGG GCT CGG AGA TGT GTA TAA GAG ACA GGA CTA CHV GGG TAT CTA ATC C-3′) with Illumina adaptor overhang sequence labeling in bold and an amplicon size of about 550 bp. The amplified DNA size was checked using a Fragment Analyzer (Agilent Technologies, Inc., Santa Clara, CA, USA). Sequencing was carried out using an Illumina Miseq platform. DNA samples were attached with indices and Illumina sequencing adapters using a Nextera XT Index Kit (Nextera XT DNA Library Preparation Kit, Illumina, San Diego, CA, USA). After library construction (amplicon size about 630 bp), the samples were mixed with MiSeq Reagent Kit v3 (600-cycle) at a final concentration of 20 pM, loaded onto a Miseq cartridge, and then onto the instrument. Sequences were binned into operational taxonomic units (OTUs) using QIIME2 (2020.11) and matched with the Greengenes database (v.13.8). From Greengenes, data were extracted on genus level, and a total of 392 genera were identified. Some genera are presented within hard brackets, which indicate a proposed taxonomy by the Greengenes database. Chao1, ACE, Fisher, and Shannon indices were chosen to characterize alpha sample diversity. For beta diversity estimation, weighted UniFrac measures were used [22]. OTUs that differed between treatments were selected based on linear discriminant analysis (LDA) effect size (LEfSe) and an LDA score above 3.0 for further analysis [23]. A *p* value less than 0.05 was considered statistically significant.

### 2.3. Bioinformatics Analysis

Bioinformatic analysis was performed using the CLC Microbial Genomics Module. Alpha diversity was measured using Shannon index (richness and evenness), Chao1 (OTU richness), ACE (OTU richness including less than 10 reads), and Fisher (relationship between the numbers of species and individuals) methods, which calculate the overall diversity of each group including the number of observed species (richness) and the evenness of observed taxonomy. Beta diversity was measured using PCoA-Weighted UniFrac, which determines the difference in microbial composition between groups. Hierarchical clustering of the top 25 OTU taxonomic abundances was performed using a heatmap to determine patterns between groups. An OTU table was generated using the CLC Microbial Genomics Module and further analyzed using LEfSe for core microbiota analysis and Phylogenetic Investigation of Communities by Reconstruction of Unobserved States 2 (PICRUSt 2) analysis for functional pathway analysis. LEfSe was performed using the Galaxy/HutLab website (http://huttenhower.sph.harvard.edu/galaxy/, accessed on 24 July 2014) to identify specific microbial markers between groups with an alpha value for the factorial Kruskal–Wallis test/pairwise Wilcoxon test of 0.05 and LDA score cutoff of 2.0. PICRUSt 2 prediction was performed using the Galaxy website according to the Kyoto Encyclopedia of Genes and Genomes (KEGG) functional pathways database and analyzed using Statistical Analysis of Metagenomic Profiles (STAMP) software. The STAMP criteria were set up with the removal of unclassified reads, *p* < 0.01, and effect size of 0.2. The results revealed a significantly different abundance in functional pathways at level 3 between groups. Spearman’s correlation and principal component analysis (PCA) in R language software (v4.0.2) were used where appropriate. 

### 2.4. Statistical Analysis

Statistical analysis was performed using IBM SPSS software (v25, IBM SPSS, Inc., Chicago, IL, USA) and GraphPad Prism 8 (v8.2.1, GraphPad software, San Diego, CA, USA). Comparisons between groups were performed using two-tailed t tests. A multivariate regression model was used to assess the risk of NSCLC with adjustments for age, sex, smoking status, and medical history (i.e., hypertension, chronic obstructive pulmonary diseases (COPD), and diabetes). Areas under the curve (AUCs) for specific microbial candidates were analyzed using receiver operating characteristic (ROC) curves. A *p* value < 0.05 was considered to be statistically significant. 

## 3. Results

### 3.1. Microbial Diversity in the NSCLC Patients and Controls

To investigate the involvement of the gut–lung axis in the risk of lung cancer, fecal microbiota from the treatment-naïve NSCLC patients (*n* = 34; age, 64.5 ± 8.9 years; males, *n* = 20 (58.8%); females, *n* = 14 (42.25%)) and normal controls (*n* = 268; age, 64.1 ± 5.9 years; males, *n* = 113 (42.2%); females, *n* = 155 (57.8%)) were analyzed after matching for age between the two groups. In addition, those with factors that could affect the gut microbiota (i.e., type 2 diabetes, hypertension, cardiovascular diseases, and associated comorbidities) were recorded. Clinical information of the NSCLC patients (i.e., age, sex, smoking history, genetic background, and disease status) are summarized in Table 1. To identify the microbial communities, the V3–V4 regions of the 16S ribosomal RNA (rRNA) bacterial gene were sequenced using the Illumina MiSeq platform. After quality filtering and contaminant removal, roughly 100,000 quality sequences per case were retained for OTU (I) clustering and downstream analysis.

To explore whether there were changes in the gut microbiota of the NSCLC patients, the commonly used Shannon diversity index (including richness and evenness) was first used to analyze the alpha diversity of the microbial composition. The results showed that the alpha diversity was decreased in the NSCLC patients, but without statistical significance (*p* = 0.087, Figure 1A). However, the Chao1, ACE, and Fisher methods showed that the alpha diversity was significantly decreased in the NSCLC patients compared with the controls (*p* = 0.016, 0.032, and 0.005, respectively, Appendix A). Furthermore, the beta diversity was significantly different between the NSCLC patients and controls (*p* < 0.001, Figure 1B). 

### 3.2. The Core Gut Microbiome in the NSCLC Patients

To identify critical gut microbes associated with the risk of lung cancer, the top 10 relative abundances at the phylum (Figure 2A) and genus (Figure 2B) levels were analyzed. The results showed that *Bacteroidetes* at the phylum level was significantly increased in the NSCLC patients (*p* = 0.039, Appendix A). At the genus level, *Lactobacillus* and *Oribacterium* spp. were significantly increased in the NSCLC patients (*p* = 0.035 and 0.002, respectively, Appendix A), whereas *Coprococcus* spp. was decreased in the NSCLC patients (*p* = 0.049, Appendix A). The relative abundances of the gut microbes at the genus level in the NSCLC patients were further analyzed using a heatmap (Figure 2C). A total of 11 bacterial species were identified in the heatmap, all of which were significantly enriched in the NSCLC patients compared with the controls (Appendix A). 

The taxonomic abundance table for the core bacteria using LEfSe analysis (LDA score >3) is illustrated in Figure 3. Both *Bacteroides caccae* and *Anaerotruncus* spp. were significantly increased in the patients with NSCLC. LEfSe analysis (Figure 3A) showed that the NSCLC patients had significantly increased abundances of *Parabacteroides distasonis*, *Anaerotruncus* spp., *Schwartzia* spp., *Morgenella* spp., *Bacteroides caccae*, *Clostridium hathewayi*, *Clostridium symbiosum*, and *Eubacterium dolichum* (*p* < 0.001, <0.001, 0.005, 0.014, <0.001, <0.001, 0.005, and *p* < 0.001, respectively, Figure 3B). The abundances of *Coprococcus* spp. and *Roseburia faecis* were significantly decreased in the NSCLC patients compared with the normal controls (*p* = 0.049 and 0.019, respectively, Figure 3C).

Smoking has been reported to be an initiating factor in lung cancer development. In addition, metabolic disorders and COPD have also been reported to influence the composition of the gut microbiota and to play a role in the development of lung cancer [24,25,26,27]. Accordingly, the abundances of *Parabacteroides distasonis* (nonsmokers, *p* < 0.001; smokers, *p* = 0.002, Figure 4), *Anaerotruncus* (nonsmokers, *p* < 0.001; smokers, *p* = 0.005, Figure 4), *Bacteroides caccae* (nonsmokers, *p* < 0.001; smokers, *p* = 0.018, Figure 4), and *Clostridium hathewayi* (nonsmokers, *p* < 0.001; smokers, *p* = 0.037, Figure 4) were significantly increased in the NSCLC patients regardless of their smoking status (Figure 4). Furthermore, the abundances of *Prevotella* spp. (*p* = 0.025) and *Coprococcus* spp. (*p* = 0.030) were reduced, but those of *Morganella* spp., *Clostridium symbiosum*, and *Eubacterium dolichum* were increased in the NSCLC patients compared with the control group (*p* = 0.044, <0.001, and <0.001, respectively, Figure 4). At the species level, there were no significant differences among the NSCLC patients, regardless of the presence of metabolic disorders (i.e., diabetes mellitus, hypertension, cardiovascular diseases; *n* = 12, Appendix A) or COPD (*n* = 5, Appendix A). We also found that these three species of bacteria were not affected by metabolic disorders or COPD (Appendix A). 

### 3.3. The Risk-Associated Gut Microbes and Related Functional Pathways in the NSCLC Patients

To identify which microbial biomarkers were associated with the risk of NSCLC, multivariate regression analysis was performed. The results revealed that *Anaerotruncus* spp. and *Bacteroides caccae* were related to the risk of NSCLC after adjusting for sex, smoking, hypertension, diabetes mellitus, COPD, and 11 core microbes (*p* = 0.003 and 0.007, respectively, Table 2). Furthermore, the AUCs derived from ROC curves were 71.4% and 66.9% for *Anaerotruncus* spp. and *Bacteroides caccae*, respectively, using the respective values of 0.019 and 0.006 (both *p* < 0.001, Figure 5). The AUC did not improve when combining these two bacteria in ROC analysis (Figure 5C). However, random forest analysis showed that *Bacteroides caccae* and *Anaerotruncus* spp. were most strongly associated with the risk of NSCLC (Appendix A, first two rows).

To explore how these two bacteria contribute to the development of NSCLC through related functional pathways, PICRUSt 2 analysis based on the KEGG pathways database was used (Figure 6A). The results showed that 12 signaling pathways were significantly correlated with the abundances of these two bacteria, as determined by Spearman’s correlation analysis (Figure 6B), and showed significant differences between the NSCLC patients and controls (Figure 6B). Steroid hormone biosynthesis, apoptosis, N-glycan biosynthesis, glycosaminoglycan degradation, lipoic acid metabolism, biosynthesis of siderophore non-ribosomal peptide, and peroxisome were significantly increased (*p* < 0.001, 0.020, <0.001, <0.001, <0.001, and 0.002, respectively, Figure 6C), whereas beta-lactam resistance, glycerolipid metabolism, chloroalkane and chloroalkene degradation, the sulfur relay system, and biosynthesis of ansamycin were significantly decreased in the NSCLC patients (*p* = 0.017, 0.006, <0.001, 0.004, and 0.009, respectively, Figure 6C). The abundance of *Bacteroides caccae* was positively correlated with steroid hormone biosynthesis, N-glycan biosynthesis, glycosaminoglycan degradation, lipoic acid metabolism, apoptosis, and peroxisome (*p* < 0.001, 0.023, <0.001, 0.039, <0.001, and <0.001, respectively, Figure 6D), but was inversely correlated with glycerolipid metabolism (*p* < 0.001, Figure 6D). *Anaerotruncus* spp. was positively and only correlated with the biosynthesis of ansamycin. No overlapping signaling pathways were modulated by *Bacteroides caccae* and *Anaerotruncus* spp. (Figure 6E).

## 4. Discussion

Recent studies have reported differences in the gut microbiota of patients with lung cancer [28,29]. Even though these studies have provided notable examples of pathogenic microbiota capable of promoting oncogenesis and regulating immune cells and the efficacy of cancer therapy, no strong bacterial oncogenic drivers have been identified, and a consensus on the underlying mechanisms or interactions has yet to be reached [3,29,30,31]. The fecal microbiome is highly dynamic and influenced by factors including age, sex, probiotics, comorbid diseases, host genetics, and certain medications such as anti-acid agents. After adjusting for potential confounding factors and comorbidities reported in previous studies [3,31], multivariate regression analysis confirmed the association of specific gut microbial biomarkers with the risk of lung cancer. We found that *Anaerotruncus* spp. and *Bacteroides caccae* were abundant gut microbes in the treatment-naïve NSCLC patients, and that they could potentially serve as predictive biomarkers for the risk of NSCLC. This is consistent with previous studies which have also reported the enrichment of *Anaerotruncus* spp. and *Bacteroides caccae* in lung cancer patients [28,29,30,32]. Intriguingly, *Clostridium symbiosum* is significantly enriched in people with lung cancer who are “non-smokers” in comparison to their composition in lung cancer patients who are classified as “smokers”. Significant stepwise increase in *C. symbiosum* abundance has been found in CRA, early CRC, and advanced CRC [4]. In addition, *C. symbiosum* colonization from coronary artery disease (CAD) patients in mice modulated the secondary bile acids pool, potentially upregulating a systemic IFN-γ response, pro-inflammatory factor production, and the Th17/Treg cell ratio [3]. Studies by Zhou et al. [5] have shown that there is a significant increase in Tregs expression and a decrease in Th17 cells in the peripheral blood of NSCLC patients compared to that of healthy patients. In particular, the Th17/Treg ratio is negatively correlated with the TNM stages [5,6]. Therefore, a relative decrease in lung cancer patients who are classified as smokers compared to non-smokers might mean a lower Th7/Treg ratio, which might contribute to advanced development. Further investigations of their interplay are required. 

The biosynthesis of ansamycin was the only and slightly enriched pathway with *Anaerotruncus* spp., which has previously been reported in patients with lung [33], gastric, and ovarian cancers [34,35]. Ansamycins such as rifamycin, ansamitocin, and geldanamycin are an important class of polyketide natural products. Ansamycin antibiotics [36,37] include important antibacterial agents, such as rifamycin, and anticancer agents such as ansamitocin [38]. Previous studies have shown high expressions of Hsp90 in lung cancer specimens and that this is associated with a poor survival rate and lymphatic metastasis in lung cancer patients [39,40,41,42], indicating that the upregulation of Hsp90 can potentially facilitate the proliferation and metastasis of lung cancer. Ansamycins have also been shown to be able to bind to a conserved pocket in the NH2-terminal adenosine triphosphate (ATP)-binding domain of Hsp90, inhibiting its activity [43]. A positive correlation between the biosynthesis of ansamycin and *Anaerotruncus* spp. may represent a compensatory impact on lung cancer formation. However, the biosynthesis of ansamycin was not significantly different between the lung cancer and normal groups in this study. Although the role of *Anaerotruncus* spp. in the development of lung cancer requires further investigation, the enrichment of *Anaerotruncus colihominis* has been found in PD-1 blockade non-responders [44]. Moreover, an in vivo study by Faith et al. using mice mono-colonized by specific bacterial strains indicated that some *Bacteroides* species, one *Parabacteroides* species, and one *Escherichia* species could significantly increase levels of regulatory T (Treg) cells [45]. In addition, a prospective study collected microbiome samples from 78 patients with NSCLC or renal cell carcinoma and found that abundances of gut microbiomes including Bacteroides caccae were associated with a longer progression-free survival with anti-PD-1 treatment. Moreover, an increased level of *Anaerotruncus* spp. was associated with worse progression-free survival [15]. These results provide crucial evidence to support the role of *Anaerotruncus* spp. in regulating host immune cells, and a possible role in the risk of NSCLC. In this study, the abundance of *Bacteroides caccae* was positively correlated with steroid hormone biosynthesis, N-glycan biosynthesis, glycosaminoglycan degradation, lipoic acid metabolism, apoptosis, and peroxisome. The effects of sex steroid hormones on the risk of lung cancer explain the sex differences in the incidence of lung cancer. Increasing epidemiological evidence has shown that increased exposure to sex steroid hormones (according to age at menarche, age at menopause, parity, and hormone use) plays a role in the development of lung cancer in women, even though the findings remain generally inconsistent [46,47,48]. Glycosylation is an enzymatic process in which carbohydrate chains called glycans are conjugated to target molecules, typically proteins and lipids [49,50]. Aberrant protein glycosylation has been demonstrated in malignant tumors, including lung cancer [51]. The positive association of enriched *Bacteroides caccae* with increased N-glycan biosynthesis and glycosaminoglycan degradation may be due to aberrations in enzymatic substrate leading to the development of lung cancer. In the present study, glycerolipid (e.g., triglycerides) metabolism was decreased and the only negatively correlated functional pathway with the abundance of *Bacteroides caccae* in the NSCLC patients. Decreased glycerolipid metabolism may result in high serum triglyceride levels, and this was associated with an increased risk of lung cancer in a large cohort study [52]. Taken together, these findings suggest that *Bacteroides caccae* may be associated with certain pathways involved in lipid metabolism, including steroid hormone biosynthesis, lipoic acid metabolism, and glycerolipid metabolism. Lipid metabolism occurs as a network of pathways with flexibility, feedback loops, and crosstalk which may increase metabolic requirements in cancer cells. Cancer cells generate many metabolic intermediates which can be used in anabolic processes for membrane building blocks or as extra- or intracellular signaling molecules to activate oncogenic cascades, eventually leading to tumor malignant progression [53,54,55]. Taken together, the association of *Bacteroides caccae* with the risk of NSCLC may be through interplay with lipid-related metabolism.

There are several limitations to this study. First, the number of lung cancer patients was limited, and our results need to be further validated in a larger cohort study. Second, although the gut microbiota of our patients was obtained at the time of diagnosis (treatment naïve), a single time point of bacterial detection may not reflect the entire range of bacterial communities in lung cancer patients. Dynamic monitoring in a longitudinal study could help to better understand the changes in gut microbiota and associations with lung cancer development and treatment. Third, microbial dysbiosis in different body fluids such as saliva or bronchoalveolar fluid has also been shown to play a crucial role in lung cancer [56]. However, we did not investigate microbial dysbiosis in different body fluids in this study. Further analysis of microbial dysbiosis as well as the lipid-related profile in different body fluids may improve our understanding of the gut–lung axis in lung cancer development and treatment. 

## 5. Conclusions

We assessed compositional changes in the gut microbiota between age-matched NSCLC patients and healthy subjects. Analysis of the associated KEGG pathways revealed cross-link between gut dysbiosis and related mechanisms of oncogenesis in the NSCLC patients. The identification of these gut microbes and associated signaling pathways may provide insights into the underlying mechanisms, which could facilitate the clinical development of biotherapeutic approaches including dietary interventions with probiotics, therapeutic administration of bacterial species or their metabolites, and selective antibiotic therapy or fecal microbial transplantation.

## Figures and Tables

**Figure 1 ijerph-19-15991-f001:**
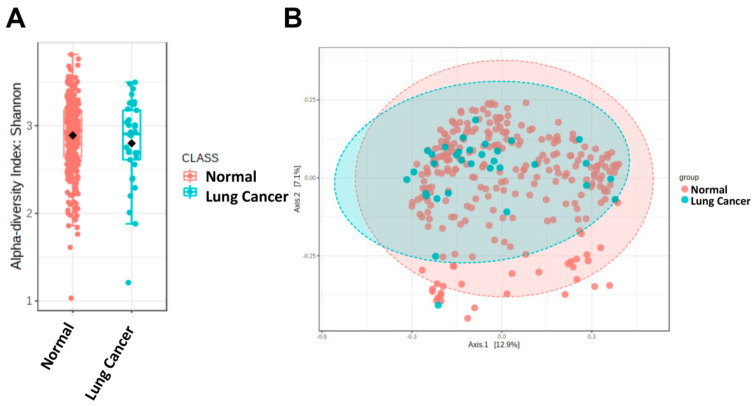
Gut dysbiosis in the patients with NSCLC. The alpha diversity in the NSCLC cases and healthy controls is illustrated using (**A**) Shannon diversity and (**B**) principal coordinates analysis (PCoA) derived from unweighted and weighted analysis of two populations (*p* < 0.0001 by PERMANOVA).

**Figure 2 ijerph-19-15991-f002:**
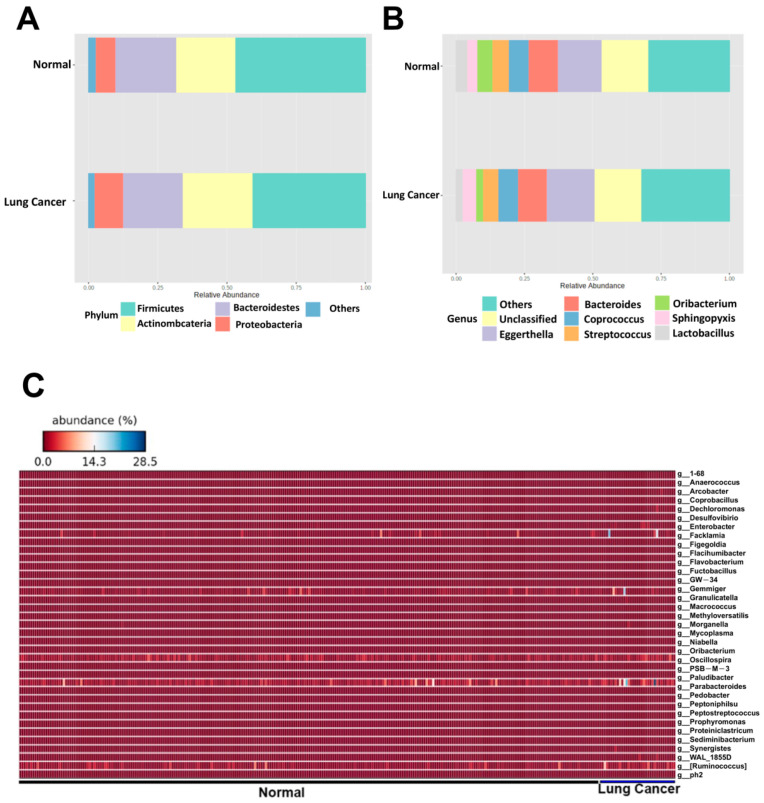
The top 10 most abundant gut microbes in patients with NSCLC are illustrated at phylum (**A**) and genus (**B**) levels. Heatmap analysis of the species level (**C**) of the gut microbes between groups.

**Figure 3 ijerph-19-15991-f003:**
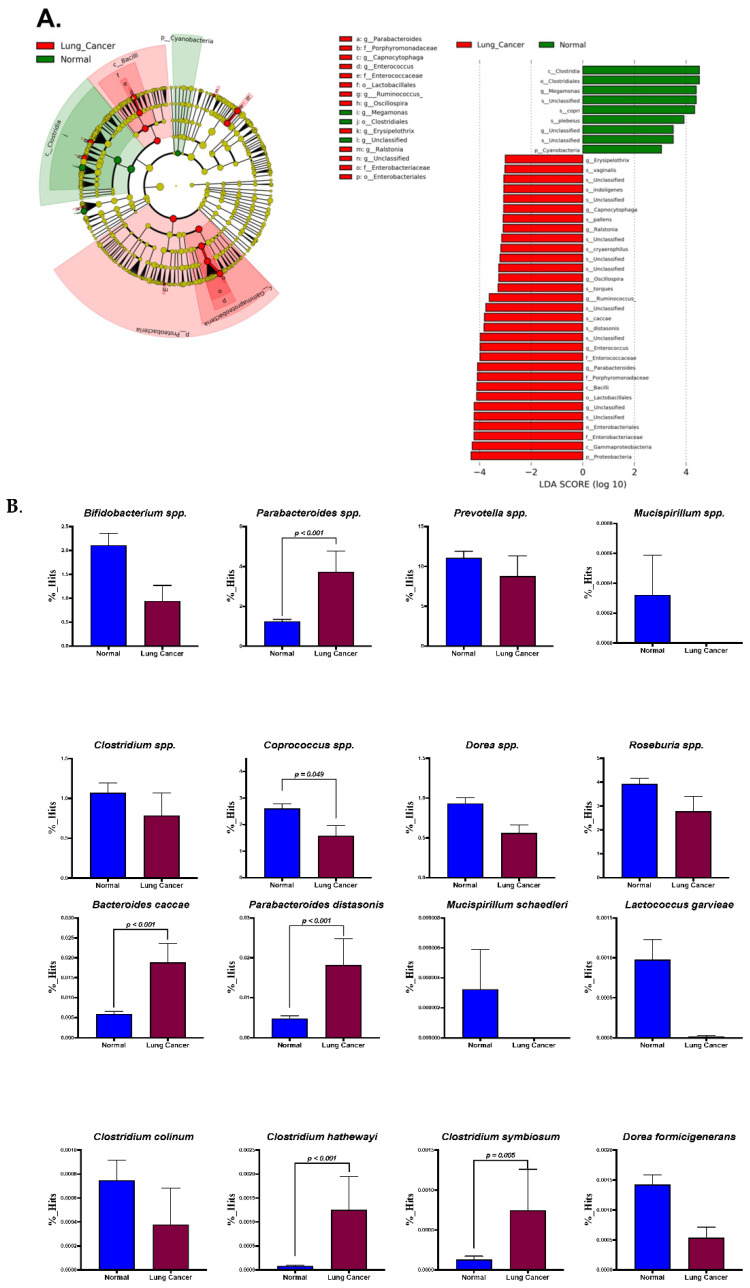
Cladogram and LEfSe plots (**A**) of the core gut microbes are separately illustrated. Bar plots based on the genus and species levels of the core microbes determined from LEfSe analysis are shown in (**B**,**C**), respectively.

**Figure 4 ijerph-19-15991-f004:**
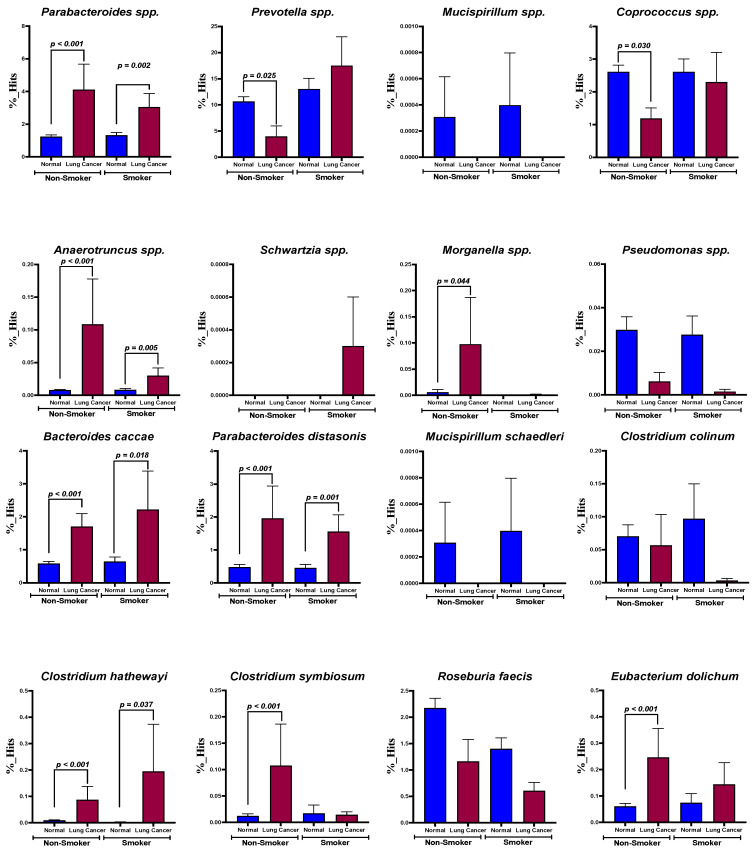
Comparisons of the abundances of core gut microbes in smokers vs. non-smokers.

**Figure 5 ijerph-19-15991-f005:**
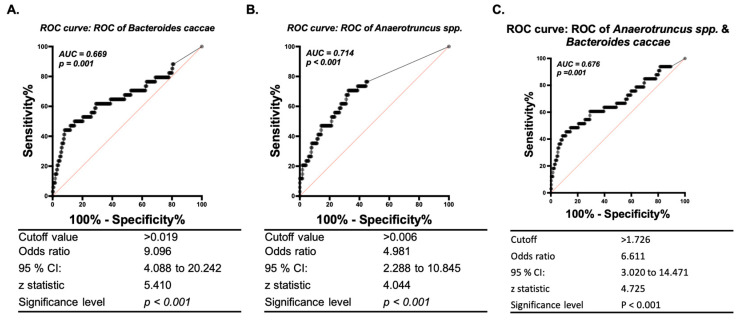
*Bacteroides caccae* and *Anaerotruncus* spp. may serve as predictive microbial biomarkers for the risk of NSCLC. The cutoff value in receiver operating characteristic (ROC) curve analysis was 0.8 including only good (0.8≤, area under the curve (AUC) < 0.9) and excellent (AUC ≥ 0.9) biosignatures. The AUCs of *Bacteroides caccae* (**A**), *Anaerotruncus* spp. (**B**), and the combination of the abundances of both bacteria (**C**) in predicting NSCLC risk were analyzed separately.

**Figure 6 ijerph-19-15991-f006:**
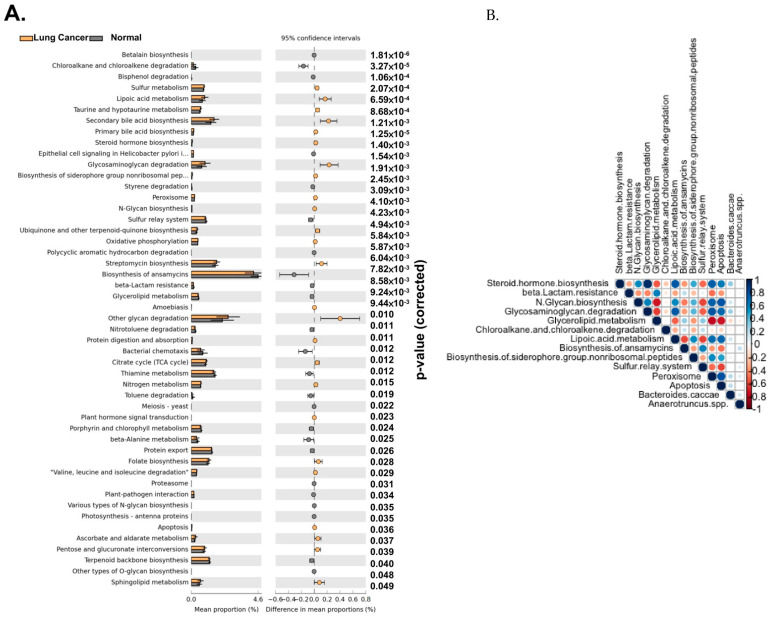
The correlation of *Bacteroides caccae* and *Anaerotruncus* spp. with specific functional pathways may contribute to the risk of NSCLC. Functional pathways were analyzed using PICRUSt 2 analysis and plotted using STAMP (v2.1.3) (**A**). Spearman’s correlation analysis (**B**) showed that 8 signaling pathways were significantly associated with *Bacteroides caccae* and *Anaerotruncus* spp. in the patients with NSCLC (**C**). Associated dot plot of the abundances of *Anaerotruncus* spp. and *Bacteroides caccae* with functional pathways in the NSCLC patients (**D**). Venn plot of *Anaerotruncus* spp.- and *Bacteroides caccae*-associated pathways showed no overlapping (**E**).

**Table 1 ijerph-19-15991-t001:** Demographic characteristics of the lung cancer patients and normal controls.

	Normal (*N* = 268)	Lung Cancer (*N* = 34)
Sex, *n* (%)		
Male	113 (42.1)	20 (58.8)
Female	155 (58.9)	14 (41.2)
Age (years)	64.1 ± 5.9	64.5 ± 8.9
Smoking, *n* (%)		
No	223 (83.2)	23 (64.7)
Yes	45 (16.8)	11 (35.3)
Current	22 (8.2)	5 (14.7)
Former	23 (8.6)	6 (17.6)
Lung cancer, *n* (%)		
Non-Small-Cell Lung Cancer		
Adenocarcinoma		26 (76.5)
Squamous		3 (8.8)
Mixed		2 (5.9)
Others		3 (8.8)
Stage, *n* (%)		
I		2 (5.9)
II		0 (0)
III		5 (14.7)
IV		27 (79.4)
EGFR Mutation, *n* (%)		
Exon 19 (del)		7 (20.6)
Exon 21 (L858R)		9 (26.5)
Mixed *		2 (5.9)
Non-detected		16 (47.0)

* Mixed: Exon19: Deletion & Exon 20: T790M: *n* = 1; Exon 18: G719X & Exon 21: L861Q: *n* = 1.

**Table 2 ijerph-19-15991-t002:** Multivariate regression analysis for lung cancer risk.

Factors	S.E.	*p* Value	Exp(B)/Odds Ratio	95% CI	
Sex (Male/Female)	0.669	0.843	1.141	0.307	4.239
Smoking (Never/Former/Current)	0.414	0.090	2.016	0.895	4.54
Hypertension (No/Yes)	9501.83	0.998	10,591,695,380	0	.
COPD (No/Yes)	11,322.247	0.998	718,2119,898	0	.
DM (No/Yes)	9565.13	0.998	1,141,677,983	0	.
*Parabacteroides* spp.	0.275	0.654	1.131	0.66	1.94
*Coprococcus* spp.	0.104	0.920	0.99	0.807	1.214
*Anaerotruncus* spp.	10.741	0.003	6.25588 × 10^13^	45,003.321	8.69625 × 10^22^
*Morganella* spp.	1.358	0.257	4.667	0.326	66.826
*Bacteroides caccae*	0.172	0.007	1.586	1.132	2.222
*Parabacteroides distasonis*	0.363	0.898	1.048	0.514	2.134
*Clostridium hathewayi*	6.825	0.560	53.513	0	34,524,785.61
*Clostridium symbiosum*	3.882	0.904	1.598	0.001	3221.089
*Roseburia faecis*	0.18	0.561	0.900	0.632	1.282
*Eubacterium dolichum*	1.673	0.853	1.364	0.051	36.222

## Data Availability

The datasets used and analyzed in the current study are available at: https://www.scidb.cn/s/ZfUFru, accessed on 29 June 2022, DOI:10.57760/sciencedb.01897.

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
