# Peer review of "Impact of Gut Dysbiosis on the Risk of Non-Small-Cell Lung Cancer"

_ijerph, 2022, doi:10.3390/ijerph192315991_

Round 1

Reviewer 1 Report

The findings of the present work were interesting. Some comments are given below for revision.

1. Line 91-101, are there the same confounders used in data adjustment in NSCLC patients and healthy subjects? Why they are chosen as confounders? Please include related information in the main text.

2. Figures 1 to 3 should be revised, because many words are not clear to read.

3. Line 456-465, some of the limitations were suggested to be improved in the present study.

Author Response

  1. Line 91-101, are there the same confounders used in data adjustment in NSCLC patients and healthy subjects? Why they are chosen as confounders? Please include related information in the main text.

Reply: Thank you for your comment. We have included relevant information and reference in the main text (Line 405). In addition, most cofounders which associate gut dysbiosis have been excluded in the healthy subjects of our study (Line 98).

  1. Figures 1 to 3 should be revised, because many words are not clear to read.

Reply: Thank you for your comment. We have revised the figures as well as the words in the text to make it clearer to read.

  1. Line 456-465, some of the limitations were suggested to be improved in the present study.

Reply: Thank you for your comment. We have amended the limitations in the text (lines 478-488).

Reviewer 2 Report

I think despite its small sample size, this is a simplistic yet important study since it narrows down specific bacterial species that are enriched in people with lung cancer, and finds metabolic associations as to explain how they may contribute to cancer progression. I only have minor comments.

1. In Figure 4, Clostridium symbiosum is significantly enriched in people with lung cancer who are "non-smokers" in comparison to their composition in lung cancer patients who are classified as "smokers". This is an intersting and unusual observation, and the possible reasons for this, if known/reported should be discussed in the discussion section.

2. Page 3, line 113, degree symbol of 20 degree celsius should be superscribed.  

3. All bacterial taxa nomenclatures throughout the paper must be italicized.

4. In Figure S4, why are the differences in species abundance for listed species not graphed separately for each of the individual metabolic disorders, ie, diabetes, cardiovascular disorders and hypertension? It would make your result point stronger if done this way.

Author Response

  1. In Figure 4, Clostridium symbiosum is significantly enriched in people with lung cancer who are "non-smokers" in comparison to their composition in lung cancer patients who are classified as "smokers". This is an interesting and unusual observation, and the possible reasons for this, if known/reported should be discussed in the discussion section.

Reply: Thank you for your comment. We have added the relevant discussion in the discussion section (lines 411-424).

  1. Page 3, line 113, degree symbol of 20 degree celsius should be superscribed.  

Reply: Thank you for your comment. We have amended the words accordingly (line 112).

  1. All bacterial taxa nomenclatures throughout the paper must be italicized.

Reply: Thank you for your comment. We have italicized all bacterial taxa nomenclatures throughout the paper.

  1. In Figure S4, why are the differences in species abundance for listed species not graphed separately for each of the individual metabolic disorders, ie, diabetes, cardiovascular disorders and hypertension? It would make your result point stronger if done this way.

Reply: We had separately graphed the listed bacteria in our study for diabetes (n=4) and hypertension (n=8).